# Greater Protection of Lower Dietary Carbohydrate to Fiber Ratio (CFR) against Poor Blood Pressure Control in Patients with Essential Hypertension: A Cross-Sectional Study

**DOI:** 10.3390/nu14214443

**Published:** 2022-10-22

**Authors:** Qingqing Dong, Lili Wang, Hanbing Hu, Lingling Cui, Anping Lu, Chunya Qian, Xiaohua Wang, Xiaojiao Du

**Affiliations:** 1Cardiology Division, The First Affiliated Hospital of Soochow University, Suzhou 215006, China; 2Department of Urology, The First Affiliated Hospital of Soochow University, Suzhou 215006, China; 3School of Nursing, Medical College, Soochow University, Suzhou 215006, China; 4Department of Orthopedics, Shanghai Jiaotong University Affiliated Sixth People’s Hospital, Shanghai 200233, China

**Keywords:** dietary carbohydrate to fiber ratio, blood pressure control, essential hypertension

## Abstract

(1) Background: Carbohydrate combined with dietary fiber (DF) applied as a surrogate marker of overall carbohydrate quality is a more essential determinant of cardiometabolic health. However, to date, no studies have applied this metric to analyze its associations with poor blood pressure control in hypertensive patients. (2) Methods: A cross-sectional design was implemented in one tertiary hospital and one community hospital in China. Using Feihua Nutrition Software to analyze participants’ two-day dietary log, the quantity of carbohydrate and fiber was obtained and the carbohydrate to fiber ratio (CFR) was calculated. The participants were divided into Q1, Q2, Q3, and Q4 groups by quartile method, from low to high according to CFR. The poor systolic and diastolic blood pressure (SBP and DBP) controls were defined as ≥140 mmHg and ≥90 mmHg, respectively. (3) Results: A convenience sample of 459 participants was included and the mean CFR was 29.6. Taking Q1 as reference, after adjusting for covariates, the CFR in Q4 was associated with higher poor SBP-controlled rate (OR, 4.374; 95% CI, 2.236–8.559). Taking Q2 as reference, after adjusting for covariates, the CFRs in Q3 and Q4 were associated with higher poor DBP-controlled rates [(OR = 1.964, 95% CI: 1.016–3.795) and (OR = 4.219, 95% CI: 2.132–8.637), respectively]. The CFR was the stronger protective determinant of SBP and DBP than DF or carbohydrate alone. (4) Conclusions: A higher CFR is a stronger risk factor for blood pressure (BP) control, and low CFR foods or a combination of corresponding food components, should be recommended in the dietary management of hypertensive patients.

## 1. Introduction

Lifestyle changes are the cornerstone of cardiovascular disease (CVD) prevention and diet is one of the most effective strategies for blood pressure (BP) control [1]. In the past decade or so, researchers have focused their attention on low salt intake, which is the most effective lifestyle for lowering BP [2]. However, the European Society of Cardiology/European Society of Hypertension guidelines report that salt hidden in processed foods accounts for 80% of total salt intake, so it is not always easy to limit salt [3]. Therefore, there is growing interest in the potential for nutrients other than sodium to improve BP control.

Carbohydrate, as an important element of diet, is important source of energy and accounts for more than half of daily caloric intake [4,5]. Compared to the lowest quintile group, the highest quintile of carbohydrate intake (g/day) had a 34% lower risk of elevated BP [6]. However carbohydrate is a readily used source of energy and is also a risk factor of obesity [7], which further affects BP [8,9]. Thus, it is not enough to consider the effect of carbohydrate quantity alone on BP or cardiovascular health [10,11,12,13,14,15]. Studies show that some attributes of carbohydrate (such as whole grain, which is a carbohydrate rich in fiber) or total fiber have been applied as surrogate markers of overall carbohydrate quality [16], and a large amount of evidence shows this is a more important determinant of health than carbohydrate quantity alone [10,11,12,13,14,15].

Dietary fiber (DF) is the fraction of the edible part of plants that are resistant to digestion and absorption in the human small intestine [17] and can regulate the gut microbiota [18], thus potentially contributing to the modulation of the renin–angiotensin–aldosterone system and autonomic nerves [19]. It is an important component of the recommended vaso-protective dietary approaches [20]. A recent meta-analysis suggests that increasing dietary fiber intake reduces both systolic blood pressure (SBP) and diastolic blood pressure (DBP) [21]. A national cohort study from China, which included 12,177 adults without hypertension at baseline to assess the relationship between carbohydrate quality and new-onset hypertension, showed that a lower intake of high-quality carbohydrates (whole grains, legumes, or fruits) and a higher intake of low-quality carbohydrates (refined rice or noodles) increased the risk of hypertension [22]. However, compared to individual measures of carbohydrate or fiber, the dietary carbohydrate to fiber ratio (CFR) provides a more holistic picture of a patient’s diet [23]. A higher dietary CFR means a diet having a lower intake of whole foods, while a lower dietary CFR ratio may indicate an increased consumption of whole foods [23]. Fontanelli et al. took grain foods having a dietary CFR of ≤10:1 to identify healthy foods and determined their association with cardiometabolic protective factors [24]. However, to date, despite evidence implicating carbohydrate and fiber intake in the shared pathogenesis of hypertension [25,26], no studies have combined both indicators into one metric to investigate its relationship with BP control in patients with essential hypertension.

The hypothesis of this study is that, the poor BP-controlled rate increases in the higher CFR group, compared with that in the low CFR group, and this increase is greater than the individual contribution of carbohydrate or DF.

## 2. Materials and Methods

### 2.1. Design and Study Participants

This was a cross-sectional study to investigate the association between the dietary CFR and the BP control in patient with essential hypertension (EH). We enrolled patients by convenience sampling who met the inclusion criteria from Jinchang Community and No. 1 Affiliated Hospital of Soochow University in Suzhou, China, between September 2019 and August 2021. The study protocol was approved by the ethics committee of the First Affiliated Hospital of Soochow University on 27 September 2019 (ECSU-2019000148) and implemented in accordance with the Declaration of Helsinki.

The inclusion criteria were adults over 18 years old with proven diagnosis of EH according to the latest guideline of hypertension [27] and obtained informed consent. Patients (1) having secondary hypertension; (2) having severe comorbidities or complications, cognitive dysfunction, diarrhea, or other gastrointestinal diseases within the past one month [28]; (3) receiving psychotherapy within the past one month, or being pregnant or lactating women; or (4) participating in other research were excluded.

### 2.2. Blood Pressure

After resting for ten minutes, upper arm BP measurements were performed twice with a five-minute break between 8:30 am and 10:30 am, in a seated position with a calibrated electronic sphygmomanometer (HEM-8102A, Dalian, China) in a community service room or hospital demonstration room before taking anti-hypertensive drugs. We took the average of the two values as the final BP value.

In this study, SBP ≥ 140 mmHg was defined as poorly controlled SBP and DBP ≥ 90 mmHg was defined as poorly controlled DBP according to the Chinese definition of hypertension [27].

### 2.3. Dietary Fiber and Carbohydrate Intake

Participants were instructed by three well-trained researchers to record their two-day dietary log [29], which included the names and estimated quality of all foods consumed on one day of the weekend and one working day. The first diet log was collected by reviewing the previous day’s detailed diet after participants completed filling out their demographics. The second diet log was collected on a day of the following weekend by phone or WeChat. The amount of nutrient intake, including carbohydrate, DF, energy, protein, fat, cholesterol, calcium, potassium, and sodium, was calculated according to dietary records using Feihua Nutrition Software (V2.7.6.10, Beijing, China). The average of the two daily nutrient intakes was taken as the final intake. CFR = dietary carbohydrate (g/day)/fiber (g/day). Participants were divided into the four groups from low to high according to CFR: Q1, Q2, Q3, and Q4, using the quartile method.

### 2.4. Sociodemographic and Clinical Data

Sociodemographic data (age, sex, marital status, education, occupational status, medical payment, duration of sleep, smoking status, and alcohol drinking) and clinical data (duration of hypertension, taking anti-hypertensive drugs) were assessed by general information questionnaire. Exercise ≤3 times a week, ≤20 min each time, and continuous time ≤3 months, was considered irregular exercise; otherwise, it was considered exercise [30]. A 10-point VAS was used to assess sleep quality, and scores ≤3, 4–6, and ≥7 were considered poor, fair, and good, respectively [31]. Body height and weight were assessed by trained and quality-monitored researchers, and participants wore only light underwear and no shoes [32]. BMI is the ratio of weight (kilograms) to height (meters) squared. Constipation symptoms include infrequent bowel movements, hard stools, sensation of incomplete evacuation or blockage and, in some instances, the use of manual maneuvers to facilitate evacuation [33]. The complications of hypertension include cerebral hemorrhage, left ventricular hypertrophy, and myocardial infarction [34]. Comorbidities are diseases not related to hypertension including malignant tumors, and liver and kidney failure. The researchers helped if participants could not distinguish between complications and comorbidities. V1.0—Depression 8b and Anxiety 8a short form were used to assess depression and anxious symptoms. Both scales contain 8 items, with each item ranked from 1 to 5, and the total score for each scale is a cumulative score of 8 items [35]. The T-score was converted to the total score according to the scoring manual, with a T-score range of 37.1 to 81.1 [36]. A T score > 55 is considered depression or anxiety [35].

### 2.5. Statistical Analysis

Statistical analysis was performed using SPSS statistical software (version 25.0; SPSS, Inc, Chicago, IL, USA) and R programming language (version 4.0.2; R Foundation for Statistical Computing, Vienna, Austria). All *p*-values were 2-tailed, and a significance level of 0.05 was used.
(1)Description of demographic, clinical data, and nutrients: The categorical variables were expressed as frequencies (percentages); the continuous variables were expressed as mean ± standard deviation (SD) if they conformed to normality, or quartiles M (P_25_, P_75_) if they were skewed.(2)Comparisons of demographic, clinical data, and nutrients: For continuous variables, the independent samples T test (normal), the Mann–Whitney U test (skewed), or the analysis of variance (ANOVA) were applied. For categorical variables, Pearson’s chi-square test, Yates’ correction chi-square, or Fisher’s exact test was used.(3)Restricted cubic splines (RCS) were plotted by the ggplot2 and rms packages of R software 4.0.2, with nodes assigned at the 5th, 35th, 65th, and 95th percentiles to assess the shape of the relationship between the dietary CFR (continuous data) and BP control.(4)Binary logistic regression was used to investigate the associations between the dietary CFR, carbohydrate, or DF alone, and poorly controlled rates of SBP and DBP.

“Crude” represents the use of univariate binary logistic regression. The variables with *p* < 0.1 in Table 1 were included in the multivariate binary logistic regression. Model 1, model 2, and model 3 were adjusted for nutrients, clinical data, and demographic data, respectively.

## 3. Results

### 3.1. General Characteristics of Participants

A total of 459 EH participants were included in the analysis. The shorter duration of hypertension, not taking anti-hypertensive drugs, and having anxiety and depression were related with the poor control of both SBP and DBP. Participants with poor SBP control were more likely to be married and have constipation, lower calcium intake, and higher sodium intake, while those with poor DBP control were younger, more likely to be male, had a higher education level, worked, were more likely to smoke and drink, and had higher BMI and cholesterol intake (Table 1).

### 3.2. The Status of Carbohydrate, Dietary Fiber Intake and CFR

The average values of dietary carbohydrate and DF of total participants were 286.12 and 11.24 g/day, respectively. The average quantities of carbohydrate intake in Q1, Q2, Q3, and Q4 were 252.49 ± 91.92, 274.29 ± 66.61, 309.41 ± 77.71, and 308.49 ± 63.18 g/day, respectively; whereas those of DF were 16.02 ± 5.48, 11.91 ± 3.10, 10.18 ± 2.72, and 6.82 ± 2.21 g/day, respectively. The average value of CFR was 29.56. The dietary CFRs in Q1, Q2, Q3, and Q4 were <20.66, 20.66 to < 26.69, 26.69 to < 36.05, and ≥36.05, respectively. Regarding the increase in the dietary CFR, the carbohydrate intake increased (*F* = 15.438, *p* < 0.001), while the DF decreased (*F* = 129.740, *p* < 0.001) (Table 2 and Table 3).

### 3.3. The Dietary Carbohydrate to Fiber Ratio and the Poor BP-Controlled Rate

In this study, 187 (40.7%) patients had poor SBP control and 179 (39.0%) had poor DBP control. The highest poorly controlled SBP and DBP rates were in Q4 (60.0% and 45.2%), while the lowest poor SBP-controlled rate and poor DBP-controlled rate were found in Q1 (31.8%) and Q2 (33.6%), respectively (Table 3).

### 3.4. The Association between the Dietary Carbohydrate to Fiber Ratio and SBP Control

Unadjusted multivariate RCS analysis showed a non-linear relationship between the continuous variable CFR and the poor SBP control (Figure 1). A CFR of 26.29 was the lowest rate of poor SBP control.

First of all, the continuous variable CFR was positively correlated with poor control of SBP after adjusting for variables (OR = 1.037, 95% CI: 1.019–1.055). Secondly, in univariate logistic regression, taking Q1 as the reference group, the dietary CFR in Q4 was significantly associated with the poorly controlled SBP rate. After adjusting for nutrient intake, demographics, and clinical indicators, the dietary CFR in Q4 was still significantly associated with the poorly controlled SBP rate (OR = 4.374, 95% CI: 2.236–8.559). In addition, the association between DF or carbohydrate alone and the poor BP-controlled rate was analyzed separately. The results indicated that after adjusting for the same covariates, the higher the DF intake, the better the control of SBP (OR= 0.868, 95% CI: 0.814–0.927); carbohydrate intake was not associated with SBP control (OR = 1.001, 95% CI: 0.998–1.005) (Table 4).

### 3.5. The Association between the Dietary Carbohydrate to Fiber Ratio and DBP Control

Unadjusted multivariate RCS analysis showed a non-linear relationship between continuous variable CFR and DBP control (Figure 2). A CFR of 26.69 was the lowest rate of poor DBP control.

First, the continuous variable CFR was positively correlated with poor control of DBP after adjusting for variables (OR = 1.033, 95% CI: 1.015–1.051). Secondly, in univariate logistic regression, taking Q2 as the reference group, the dietary CFRs in remaining groups were not significantly associated with the poorly controlled DBP rate. However, after adjusting for nutrient intake, demographics, and clinical indicators, the CFRs in Q3 (OR = 1.964, 95% CI: 1.016–3.795) and Q4 (OR = 4.291, 95% CI: 2.132–8.637) were significantly associated with the poorly controlled DBP rates. In addition, the association between DF or carbohydrate alone and the poor DBP-controlled rate was analyzed separately. The results indicated that after adjusting for the same covariates, the higher the DF intake, the better the control of DBP (OR = 0.920, 95% CI: 0.869–0.974); carbohydrate intake was not associated with SBP control (OR = 1.004, 95% CI: 1.000–1.007) (Table 5).

## 4. Discussion

The dietary CFR is an indicator used to define carbohydrate quality. Higher ratios, reflective of a poorer carbohydrate quality diet, have been associated with higher risk of T2DM [12] and coronary heart disease (CHD) [13]. A previous study identified grain foods having a dietary CFR ≤ 10:1 as being healthy foods, and found that a higher intake of these foods was associated with less atherogenic dyslipidemia and insulin resistance [24]. However, the Chinese population is accustomed to treating refined rice and noodles as staple foods, resulting in less intake of whole grain foods, so their dietary whole grain fiber intake is significantly reduced; furthermore, they are used to eating more fiber-rich vegetables, which to some extent compensates for the deficiencies of insufficient grain DF intake [37]. Thus, this study took the ratio of total carbohydrates to total DF intake as a pragmatic metric, and analyzed its relationship with the rate of poor BP control in patients with hypertension. The results showed that, compared with the group having low dietary CFR, the high dietary CFR group had a significantly higher rate of poor SBP and DBP control, which is generally in line with our research hypothesis.

### 4.1. General Characteristics of Participants

The average age of the participants in this study was 51.31 ± 12.62 years and the majority were male (62.7%), which is consistent with the epidemiological characteristics of hypertension in China [27,38]. Compared to participants with good control of BP, those with poorly controlled BP were more likely to be married, smoke, drink alcohol, be constipated, have a higher BMI, and to not take anti-hypertensive drugs, which is consistent with the study by Tuoyire and Wu et al. [39,40]. Participants were also younger and employed [41], which are the risk factors for increased BP [42]. The BP control was rather poor for the highly educated, which was inconsistent with Sabine’s study [43]. The possible reason for this is that, compared to participants having a lower level of education, the more educated patients were more likely to be working (50.2% vs. 72.4%). Participants with shorter disease duration often have less awareness of hypertension, which may lead to poorer adherence to treatment [44], and further result in poor BP control.

It has been found that the anxiety and depression in hypertensive patients can lead to poor adherence to anti-hypertensive medication and, consequently, poor BP control [45], which is consistent with this study. The results of this study showed that higher dietary cholesterol was associated with poor BP control; this may be due to higher dietary cholesterol raising serum cholesterol levels [46], which are positively correlated with BP [47]. Convincing evidence shows that high dietary sodium intake and low potassium intake are associated with increased BP [48], and this study also supported this view.

### 4.2. The Status of Carbohydrate and Dietary Fiber Intake

Staple foods are mainly coarse grains in the traditional Chinese diet; therefore, DF intake increases as carbohydrate intake increases [49]. However, recently, China has experienced a shift from traditional to Western dietary patterns, with a decrease in cereal and vegetable consumption, and an increase in meat and packaged food consumption [37], which leads to significant changes in the macronutrient composition, including fiber in the diet [50]. In this study, the mean values of dietary carbohydrate, fiber, and CFR were 286.12, 11.24, and 29.56, respectively. The mean dietary carbohydrate was basically in the normal range [51], while fiber intake was significantly reduced (far lower than 25–30 g/day recommended by 2022 dietary guidelines for Chinese residents). The statuses of dietary carbohydrate and fiber intake (Table 2) indicate that, with the increase in the dietary CFR, the carbohydrate intake increased, while the DF intake decreased; this means the factors contributing to the change in dietary CFR include the changes in both carbohydrate and DF intake. At the same time, the proportion of the population having a dietary CFR < 20.66 was only 25%, which is far more than 10:1 C/F ratio [24] identified as being healthy grain foods. This indicates that low-quality carbohydrate was consumed by patients with essential hypertension.

### 4.3. Association between Dietary CFR and SBP

There has been an increased focus on carbohydrate quality over quantity in determining the risk of chronic disease [16]. However, carbohydrate quality is a multilayered concept [24]. First, both carbohydrate and fiber are mainly contained in plant-based foods, so the designation of carbohydrates and dietary fiber by food group (e.g., fruit, vegetables, and cereals) may be a useful indicator for assessing carbohydrate quality [52]. Second, dietary recommendations for carbohydrate quality are usually to increase dietary fiber and whole grains, and limit added sugars. Such recommendations ignore refined grain intake and the fact that products containing more fiber also contain more starch and sugar [24]. It has been found that dietary CFR is one of the simplest and most effective ratios used to establish carbohydrate quality [52], because it combines the relative contributions of starch and sugar with DF, and is more easily understood by the public.

Lower dietary CFR as a protective factor for T2DM and CHD has been proven by relevant studies. However, there is currently little evidence on the association between CFR and poor BP control. SBP is known to be an important independent risk factor for cardiovascular disease (CVD) in hypertensive patients over 50 years of age [53], and is the primary contributor to the global burden of disease [54]. In this study, lower CFR as a protective factor reduced the risk of poorly controlled SBP rate by 3.374 multiples (OR = 4.374; 95% CI: 2.236–8.559), compared with that in higher CFR. In addition, we found a stronger association between dietary CFR and the poor SBP-control rate than with DF (OR = 0.868; 95% CI: 0.814–0.927) alone or carbohydrate (OR, 1.001; 95% CI, 0.998–1.005) alone, after adjusting for covariates. This indicates a potential biological interaction between these nutrients could explain these findings.

### 4.4. Association between Dietary CFR and DBP

Due to the concept that “SBP is the most important”, the awareness and treatment of DBP in hypertensive patients is very low [55]; for example, according to the data of the PEACE Million People Project, in China the awareness rate is 10.3% and the untreated rate is 86.1% [56]. Concordant elevations in both SBP and DBP pose the greatest risk for cardiovascular disease-related mortality [57].

In this study, we found that compared with Q2, CFRs in Q3 (OR = 1.964; 95% CI: 1.106–3.795) and Q4 (OR = 4.291; 95% CI: 2.132–8.637) had higher poor DBP-controlled rates, after adjusting for covariates. At the same time, we found that CFRs had a stronger association than carbohydrate (OR = 1.004; 95% CI:1.000–1.007) or DF (OR = 0.920; 95% CI: 0.869–0.974) alone.

Regarding the mechanism of CFR lowering blood pressure, one area of emerging interest is intestinal dysbiosis induced by a deficit in high-glycemic/high-carbohydrate food in DF [58], which contributes to the development of hypertension. DF is fermented by the intestinal microbiota to produce short-chain fatty acids (SCFAs). Adequate DF regulates the gut microbiota ecosystem, which increases the number of bacteria that produce SCFAs and further enhances the production of SCFAs [59,60]. SCFAs activate G protein-coupled receptors (GPR43, GPR41) on kidney cells to inhibit renin secretion, resulting in lower blood pressure [61,62]. SCFAs also act on GCPR expressed in the vagal nerve ganglion, which activates the vagus nerve to lower BP [63,64]. Another mechanism may be that low carbohydrate decreases insulin resistance, which in turn reduces stimulation of endothelial function and inflammation, thus lowering BP. In addition, low carbohydrate also reduces insulin levels and further weakens sympathetic nervous system activity, thereby reducing vascular resistance and cardiac output, and promoting sodium excretion, ultimately lowering BP.

## 5. Conclusions

This study showed a lower dietary carbohydrate to fiber ratio had a greater protection against poor blood pressure control in patients with essential hypertension. Thus, low CFR foods or a combination of corresponding food components can be recommended in the dietary management of hypertensive patients.

## 6. Strengths and Limitations

To our knowledge, this is the first study to investigate the relationship between the dietary CFR and blood pressure control. We found that a low quality of carbohydrate was consumed by most participants with essential hypertension; a lower dietary CFR showed a greater protective effect against poor BP control in patients with essential hypertension. However, there are several limitations to this study that need to be considered. First, this study was a cross-sectional design and a causal relationship could not be established. Therefore, further studies are needed to assess the relationship between the CFR and BP control. Second, we used 24 h dietary recall for dietary intake assessment. Therefore, some participants may have misreported their dietary intake due to memory-related issues. Third, the effect of other micronutrients such as vitamin D and minerals on BP control was not considered, although the intakes of sodium and potassium were adjusted for in the multivariate model. Finally, this study did not consider participants’ adherence to anti-hypertensive medication, which may limit the interpretation of the effect of CFR on BP control.

## Figures and Tables

**Figure 1 nutrients-14-04443-f001:**
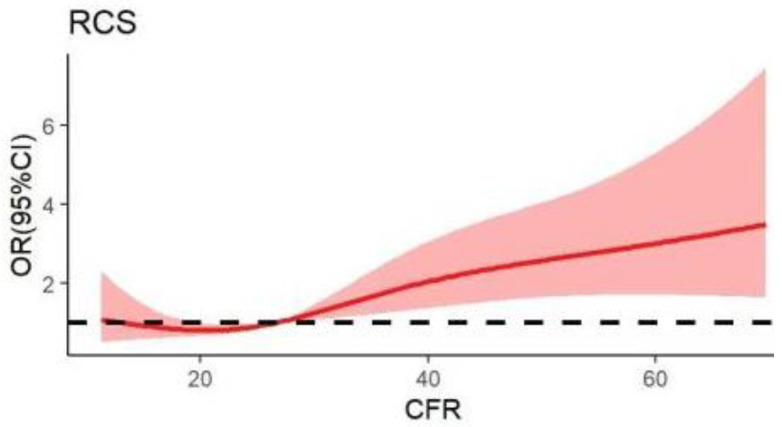
Correlation curve between the carbohydrate to dietary fiber ratio (CFR) and systolic blood pressure (SBP) control. The solid line indicates the estimated SBP control risk and the shaded area indicates the 95% confidence interval (CI).

**Figure 2 nutrients-14-04443-f002:**
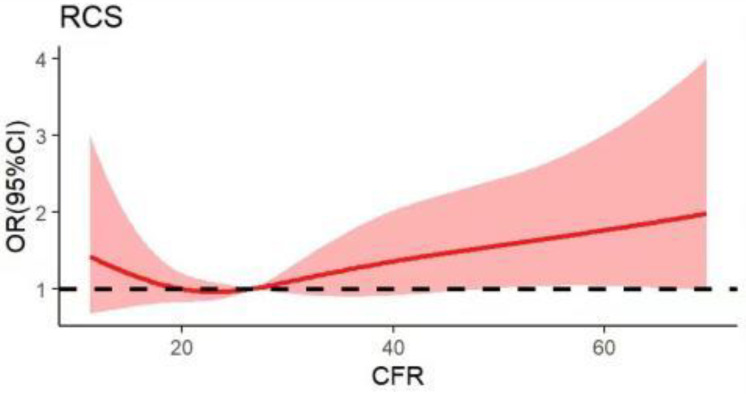
Correlation curve between the carbohydrate to dietary fiber ratio (CFR) and diastolic blood pressure (DBP) control. The solid line indicates the estimated SBP control risk and the shaded area indicates the 95% confidence interval.

**Table 1 nutrients-14-04443-t001:** Socio-demographic, clinical characteristics, and nutrients (*n* = 459).

Characteristics		SBP Control $\bar{x}$ ± s/M (P_25_,P_75_)/*n*(%)			DBP Control $\bar{x}$ ± s/M (P_25_,P_75_)/*n*(%)		
		Poor (*n* = 187)	*t/χ* ^2^ */z*	*p*	Poor (*n* = 179)	*t/χ* ^2^ */z*	*p*
Demographic Information							
Age (year)		50.09 ± 13.90	1.712 ^a^	0.088	46.15 ± 11.56	7.393 ^a^	<0.001 ***
Sex	Male	115 (61.5)	0.210 ^b^	0.647	124 (69.3)	5.350 ^b^	0.021 *
Marital status	Single	1 (0.5)	11.327 ^c^	0.001 **	5 (2.8)	1.221 ^c^	0.625
	Married	185 (98.9)			173 (96.6)		
	Others	1 (0.5)			1 (0.6)		
Education degree	High school and above	109 (58.3)	1.859 ^b^	0.173	121 (67.6)	20.402 ^b^	<0.001 ***
Occupational status	On the job	121 (64.7)	1.922 ^c^	0.393	143 (79.9)	40.333 ^c^	<0.001 ***
	Retired	62 (33.2)			34 (19.0)		
	No job	4 (2.1)			2 (1.1)		
Medical payment method	Medical Insurance	12 (6.4)	1.097 ^b^	0.578	11 (6.1)	2.135 ^b^	0.344
	Agricultural insurance	14 (7.5)			11 (6.1)		
	Self-paying	161 (86.1)			157 (87.7)		
Regular exercise	No	137 (73.3)	0.058 ^b^	0.810	131 (73.2)	0.069 ^b^	0.793
Duration of sleep (h/day)		6.82 ± 1.07	0.445 ^a^	0.657	6.90 ± 1.05	−0.923 ^a^	0.357
Quality of sleep	Good	16 (8.6)	2.099 ^b^	0.350	14 (7.8)	0.916 ^b^	0.633
	Fair	113 (60.4)			111 (62.0)		
	Poor	58 (31.0)			54 (30.2)		
Smoking status	Yes	49 (26.2)	0.041 ^b^	0.840	60 (33.5)	9.375 ^b^	0.002 **
Alcohol drinking	Yes	47 (25.1)	0.236 ^b^	0.627	59 (33.0)	13.031 ^b^	<0.001 ***
Clinical Information							
BMI (kg/m^2^)		25.15 (23.35, 27.43)	−1.757 ^d^	0.079	25.35 (23.84, 27.83)	−3.572 ^d^	<0.001 *
Constipation	Yes	25 (13.4)	5.949 ^b^	0.015 *	21 (11.7)	1.931 ^b^	0.165
Duration of HTN (year)		2.00 (0.50, 7.00)	−3.852 ^d^	<0.001 ***	2.00 (0.50, 7.00)	−3.593 ^d^	<0.001 ***
Taking drugs	Yes	130 (69.5)	11.021 ^b^	0.001 **	118 (65.9)	21.839 ^b^	<0.001 ***
Complication	Yes	4 (2.1)	1.793 ^e^	0.181	1 (0.6)	0.172 ^e^	0.678
Comorbidity	Yes	27 (14.4)	0.234 ^b^	0.629	21 (11.7)	0.792 ^b^	0.373
Anxiety	Yes	45 (24.1)	6.432 ^b^	0.011 *	44 (24.6)	7.147 ^b^	0.008 **
Depression	Yes	52 (27.8)	13.460 ^b^	<0.001 ***	54 (30.2)	20.757 ^b^	<0.001 ***
Nutrition intake (d)							
Energy (kcal)		1966.12 ± 329.44	1.597 ^a^	0.111	2005.84 ± 348.66	−0.303 ^a^	0.762
Protein (g)		72.28 ± 22.02	1.582 ^a^	0.114	76.71 ± 22.64	−1.890 ^a^	0.059
Fat (g)		69.29 ± 19.37	−0.395 ^a^	0.693	70.75 ± 19.49	−1.631 ^a^	0.104
Cholesterol (mg)		377.29 ± 226.50	−1.932 ^a^	0.054	379.70 ± 238.81	−2.056 ^a^	0.040 *
Calcium (mg)		456.66 ± 281.03	3.866 ^a^	<0.001 ***	620.03 ± 595.27	−1.416 ^a^	0.157
Potassium (mg)		1656.10 ± 483.42	1.466 ^a^	0.143	1722.64 ± 445.44	−1.138 ^a^	0.256
Sodium (mg)		2126.86 ± 631.48	−2.135 ^a^	0.033 *	2008.35 ± 843.01	0.418 ^a^	0.676

Notes: ***: *p* < 0.001; **: *p* < 0.01; *: *p* < 0.05. SBP, systolic blood pressure; DBP, diastolic blood pressure; HTN: hypertension; Taking drugs means taking antihypertensive drugs; BMI, body mass index. ^a^ Independent-samples T test; ^b^ Pearson chi-square; ^c^ Fisher’s exact test; ^d^ Mann–Whitney U; M (P_25_, P_75_), median (25th and 75th percentiles); ^e^ Yates’ correction chi-square.

**Table 2 nutrients-14-04443-t002:** The status of carbohydrate and dietary fiber intake.

Group	Carbohydrate (g/day)				Fiber (g/day)			
	Range	$\bar{X}$ *± S*	*F*	*p*	Range	$\bar{X}$ *± S*	*F*	*p*
Q1	(77.00, 559.00)	252.49 ± 91.92	15.438	<0.001 ***	(7.30, 28.85)	16.02 ± 5.48	129.740	<0.001 ***
Q2	(135.00, 454.00)	274.29 ± 66.61			(5.40, 21.60)	11.91 ± 3.10		
Q3	(149.00, 515.00)	309.41 ± 77.71			(4.50, 18.50)	10.18 ± 2.72		
Q4	(191.00, 503.00)	308.49 ± 63.18			(2.70, 11.40)	6.82 ± 2.21		

Notes: Q: Quintile; Q1, P_0_−P_25;_ Q2, P_25_−P_50_; Q3, P_50_–P_75_; Q4, P_75_–P_100_. ***: *p* < 0.001.

**Table 3 nutrients-14-04443-t003:** The rates of poor blood pressure control in different dietary carbohydrate to fiber ratios.

Carbohydrate to Fiber Ratio	SBP Control, *n*(%)			DBP Control, *n*(%)		
	Poor (*n* = 187)	*χ* ^2^	*p*	Poor (*n* = 179)	*χ* ^2^	*p*
Q1 (<20.66)	36 (31.8)	25.265	<0.001 ***	43 (38.0)	3.473	0.324
Q2 (20.66 to <26.69)	42 (36.2)			39 (33.6)		
Q3 (26.69 to <36.05)	40 (34.7)			45 (39.1)		
Q4 (≥36.05)	69 (60.0)			52 (45.2)		

Notes: Q1, P_0_−P_25_; Q2, P_25_−P_50_; Q3, P_50_–P_75_; Q4, P_75_–P_100_. ***: *p* < 0.001. SBP, systolic blood pressure; DBP, diastolic blood pressure.

**Table 4 nutrients-14-04443-t004:** Odds ratios and 95% confidence intervals for SBP according to different CFRs.

Variable		Crude	Model 1	Model 2	Model 3
		OR (95% CI)	OR (95% CI)	OR (95% CI)	OR (95% CI)
Dietary CFR		1.029 (1.015, 1.044) ***	1.027 (1.011, 1.043) **	1.037 (1.020, 1.055) ***	1.037 (1.019, 1.055) ***
Group by quartile of CFR	Q1	1.00 (Ref.)	1.00 (Ref.)	1.00 (Ref.)	1.00 (Ref.)
	Q2	1.263 (0.730, 2.182)	1.208 (0.686, 2.128)	1.212 (0.662, 2.216)	1.118 (0.602, 2.078)
	Q3	1.170 (0.675, 2.029)	1.119 (0.620, 2.021)	1.323 (0.706, 2.480)	1.224 (0.645, 2.324)
	Q4	3.365 (1.952, 5.800) ***	3.200 (1.752, 5.847) ***	4.522 (2.356, 8.679) ***	4.374 (2.236, 8.559) ***
Carbohydrate	Alone	0.998 (0.995, 1.001)	0.999 (0.996, 1.002)	1.001 (0.998, 1.004)	1.001 (0.998, 1.005)
Fiber	Alone	0.879 (0.835, 0.924) ***	0.887 (0.839, 0.939) ***	0.873 (0.822, 0.927) ***	0.868 (0.814, 0.927) ***
Covariates	Cholesterol (mg/day)	-	1.001 (1.000, 1.002) **	-	-
	Calcium (mg/day)	-	0.999 (0.999, 1.000) *	0.999 (0.999, 1.000) **	0.999 (0.999, 1.000) *
	Duration of HTN (year)	-	-	0.962 (0.932, 0.992) *	0.967 (0.936, 0.999) *
	Taking drugs	-	-	0.380 (0.219, 0.660) **	0.384 (0.221, 0.669) **
	Depression	-	-	1.969 (1.105, 3.510) *	1.861 (1.040, 3.328) *

Notes: SBP, systolic blood pressure; CFR, dietary carbohydrate to fiber ratio; ***: *p* < 0.001; **: *p* < 0.01; *: *p* < 0.05. Model 1, Adjusted for cholesterol, calcium, sodium. Model 2, Adjusted for variables in model 1 + body mass index (BMI), constipation, duration of hypertension (HTN), taking antihypertensive drugs, depression, anxiety. Model 3, Adjusted for variables in model 1 + model 2 + age, marital status.

**Table 5 nutrients-14-04443-t005:** Odds ratios and 95% confidence intervals for DBP according to different CFRs.

Variable		Crude	Model 1	Model 2	Model 3
		OR (95% CI)	OR (95% CI)	OR (95% CI)	OR (95% CI)
Dietary CFR		1.011 (0.998, 1.024)	1.016 (1.002, 1.031) *	1.027 (1.012, 1.043) **	1.033 (1.015, 1.051) ***
Group by quartile of CFR	Q2	1.00 (Ref.)	1.00 (Ref.)	1.00 (Ref.)	1.00 (Ref.)
	Q1	1.164 (0.678, 1.997)	1.133 (0.655, 1.960)	1.246 (0.695, 2.232)	1.213 (0.644, 2.283)
	Q3	1.253 (0.732, 2.145)	1.466 (0.843, 2.550)	1.962 (1.072, 3.591) *	1.964 (1.016, 3.795) *
	Q4	1.634 (0.958, 2.787)	2.070 (1.179, 3.635) *	3.179 (1.693, 5.968) ***	4.291 (2.132, 8.637) ***
Carbohydrate	Alone	0.999 (0.997, 1.002)	0.999 (0.996, 1.002)	1.001 (0.998, 1.005)	1.004 (1.000, 1.007)
Fiber	Alone	0.966 (0.925, 1.008)	0.933 (0.888, 0.980) **	0.923 (0.876, 0.973) **	0.920 (0.869, 0.974) **
Covariates	Duration of HTN (year)	-	-	0.961 (0.931, 0.991) *	-
	Taking drugs	-	-	0.408 (0.243, 0.686) **	0.493 (0.281, 0.868) *
	Depression	-	-	2.547 (1.447, 4.484) **	2.292 (1.252, 4.194) **
	Age	-	-	-	0.964 (0.942, 0.986) **
	Education degree	-	-	-	0.561 (0.335, 0.939) *
	Retired	-	-	-	0.528 (0.297, 0.939) *
	Alcohol drinking	-	-	-	2.225 (1.332, 3.716) **

Notes: DBP, diastolic blood pressure; CFR, dietary carbohydrate to fiber ratio; ***: *p* < 0.001; **: *p* < 0.01; *: *p* < 0.05. Model 1, Adjusted for protein, cholesterol. Model 2, Adjusted for variables in model 1 + body mass index (BMI), duration of hypertension (HTN), taking antihypertensive drugs, depression, anxiety. Model 3, Adjusted for variables in model 1 + model 2 + age, sex, education degree, occupational status, smoking status, alcohol drinking.

## Data Availability

The data in this study involves privacy issues, data should not be shared.

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
