# Peer review of "Greater Protection of Lower Dietary Carbohydrate to Fiber Ratio (CFR) against Poor Blood Pressure Control in Patients with Essential Hypertension: A Cross-Sectional Study"

_nutrients, 2022, doi:10.3390/nu14214443_

Round 1

Reviewer 1 Report

“The paper of Dong and colleagues evaluates the association between carbohydrates to fibre ratio with blood pressure control in hypertensive patients. They report that a higher CFR is the stronger risk factor for BP control in sample of Chinese population (n=459). The study is well conducted. The reviewer has no major concerns with the study design, conduct and the written report (except for Introduction). This study makes a good contribution to the current established understanding of the association between carbohydrates quality with NCDs including CVD, hypertension and T2D. I have minor reservations which are brought to the attention of the authors in a most constructive manner for their considerations as noted below.”

“The fact that this study was conducted in Chinese population who have a higher percentage of energy from carbohydrate compared to the non-Asian countries is interesting for me and provides a research-worthy topic. As I mentioned above, this study makes a good contribution, but not necessarily adding new findings, to the our established/proven understanding of the role of good carbohydrate quality. There are number of wide-ranging available cohort published studies that makes me thinking in what aspects this current paper is going to add to our knowledge. For example, recent comprehensive a Nationwide cohort study in China (Dietary Carbohydrate Intake and New-Onset Hypertension: A Nationwide Cohort Study in China: Hypertension. 2021; 78:422–430) provides a very complete report of this association in a sample of bigger population of 12,177 participants, compared to this current study. I highly appreciate if authors can expand (highlight) a bit more in the introduction section about the importance of their research (for instance, this study has been done on people with essential hypertension and focused on BP control, and/or, more detailed analysis on fibre intake and/or CFR ratio), and also particularly in the conclusion section about what this study adds to our current knowledge and possibly having a future direction section for the research pathway”. It seems a repetition of previous works with a slight change, if you are not justifying the importance and relevance of performing the re-analysis of this cohort population”.

“The introduction needs to be revised, as this study is focused on BP control in hypertension people, I am suggesting to provide only examples of the studies that have been done on BP control or onset of BP, and have a more focused approach in your introduction to make your readers ready that you are going to aim a particular group of people, rather than listing some studies on depressive symptoms or postprandial blood glucose which are totally irrelevant to your current research. Also, I suggest to highlight the importance of your work in the introduction, rather than simply saying “no studies have combined the two indicators into one metric to analyse the association between them”.

Author Response

请参阅附件。

Reviewer 2 Report

Dear Editor,

I carefully revised the manuscript "Greater protection of lower dietary carbohydrate to fiber ratio (CFR) against poor blood pressure control in patients with essential hypertension: a cross-sectional study" by Dong et al.

My comments and suggestions for the authors are the following:

 - Lines 93, 94: The authors should specify which questionnaire they used.

 - The limitations of the study should be further discussed.

 - English language needs to carefully revised and improved.

 - References (n. 7, 16, 30, 54) should be formatted following the Instructions for the authors of the journal.

 - The authors should consider to refer to doi: 10.1007/s40292-021-00474-6 and doi: 10.33963/KP.15468.

 - In the abstract, all the abbreviations used should be defined.

Round 2

Reviewer 2 Report

Dear Editor,

I carefully read the revised version of the manuscript, that is significantly improved in comparison with the original version.